# The Effect of Physical Exercise on People with Psychosis: A Qualitative Critical Review of Neuroimaging Findings

**DOI:** 10.3390/brainsci13060923

**Published:** 2023-06-07

**Authors:** Francesca Saviola, Giacomo Deste, Stefano Barlati, Antonio Vita, Roberto Gasparotti, Daniele Corbo

**Affiliations:** 1Department of Medical and Surgical Specialties, Radiological Sciences and Public Health, University of Brescia, 25123 Brescia, Italy; francesca.saviola@unibs.it (F.S.); stefano.barlati@unibs.it (S.B.); roberto.gasparotti@unibs.it (R.G.); 2Department of Mental Health and Addiction Services, ASST Spedali Civili of Brescia, 25123 Brescia, Italy; 3Neuroradiology Unit, ASST Spedali Civili of Brescia, 25123 Brescia, Italy

**Keywords:** physical exercise, neural plasticity, brain connectomics, psychosis, genuine movement abnormalities

## Abstract

Recently, genuine motor abnormalities have been recognized as prodromal and predictive signs of psychosis onset and progression. Therefore, physical exercise could represent a potentially relevant clinical tool in promoting the reshaping of neural connections in motor circuitry. The aim of this review is to provide an overview of the literature on neuroimaging findings as a result of physical treatment in psychosis cohorts. Twenty-one studies, all research articles, were included and discussed in this narrative review. Here, we first outlined how the psychotic brain is susceptible to structural plastic changes after aerobic physical training in pathognomic brain areas (i.e., temporal, hippocampal and parahippocampal regions). Secondly, we focused on functional changes, both region-specific and in terms of connections, to gain insights into the involvement of distant but inter-related neural regions in the plastic process occurring after treatment. Third, we attempted to bridge neural plastic changes occurring after physical interventions with clinical and cognitive outcomes of psychotic patients in order to assess the relevance of such neural reshaping in the psychiatric rehabilitation field. In conclusion, we suggest that the current state of the art is presenting physical intervention as effective in promoting neural changes for patients with psychosis; it is not only useful at the onset of the pathology but also in improving the course of the illness and its functional outcome. However, more evidence is needed to improve our knowledge of the efficacy of physical exercise in plastically reorganizing the psychotic brain in the long term, especially within regions lacking specific investigations, such as motor circuitry.

## 1. Introduction

Psychosis is a pathological condition characterized by a set of severe symptoms affecting different levels of everyday life (i.e., social, cognitive and perceptual domains) and is related to poor functional outcomes [1]. Signs of altered motor development are increasingly recognized as an important marker of risk for psychosis [2,3]. Indeed, a growing body of literature suggests that genuine movement abnormalities (GMA; e.g., akathisia, catatonia, dyskinesia, parkinsonism and psychomotor slowing) [4] are present long before the first signs of thought disorder, and prospective studies of youth at risk show that GMA may predict the transition to psychosis [5,6,7]. Therefore, it is alleged that motor circuitry has a relevant role in the pathophysiology of psychosis [4]. 

At a neural level, several brain regions were shown to be structurally implicated in the neuropathology of psychosis [8,9,10,11,12], with a main focus on the medial temporal lobe (together with hippocampal and parahippocampal areas), parietal lobe and subcortical areas involvement. However, there is still limited information about how functional motor areas disruption co-occur with GMA in psychosis and how different motor circuits may contribute to the understanding of the pathology [13].

Specifically, the functional somatomotor network (SMN) is a large-scale functional network usually involving both (i) motor circuitry within the precentral gyrus together with supplementary motor areas and (ii) somatosensory regions relying on the postcentral gyrus [14].

The SMN connections were found to be functional transdiagnostic hubs across several psychiatric disorders [15,16]. Indeed, dysconnectivity within the SMN is believed to be a common feature across multiple psychiatric conditions; whereas its connections to frontal areas are rather disrupted, with an illness-specific pattern, when considering psychotic disorders [16]. These findings highlight that psychiatric disorders share common impairments linked to perception and motor output (i.e., related to the bottom-up networking within regions of the motor circuitry); whereas unique brain connectivity profiles can be found for each pathology mediating more distinctive features of the disorder (i.e., related to the top-down networking between motor circuitry and frontal areas) [4,17]. 

Hence, in the case of psychosis, efficiently targeting motor circuitry connections could be of foremost importance in treating and acting on overall symptomatology. Clinical treatment of psychosis is generally carried out through the application of a pharmacological therapy combined with several non-pharmacological psychosocial interventions [18,19] consistently varying in types across different guidelines. Evidence-based psychosocial treatments are aimed at improving the quality of life and at a more efficient management of the illness; among them, the ones recognized as most common are cognitive-behavioral therapy, social skills training and cognitive remediation. Recently, physical exercise has also appeared as a valid tool for psychotic rehabilitation [20], not only given its potential to target GMA from the beginning of the pathology—or even in high-risk subjects with a preventive role—but also as a non-pharmacological treatment acknowledged to improve symptoms and cognitive performance in schizophrenia [21]. Furthermore, physical training is characterized by high practical feasibility, which makes it extremely appealing for psychiatric rehabilitation. Thus, a better understanding of its expected neural effects on the multifaceted symptomatology of psychosis and its potential outcomes may represent an important clinical topic. 

### Aims

Therefore, the general aim of this review is to provide an overview of brain connectomics literature (both structural and functional) on physical exercise plasticity effects as a treatment for mental illness, with a particular focus on psychosis. 

The first more specific aim is to evaluate how volumetric properties of diverse brain regions are affected by physical training, considering both areas thought to be pathogenic for the psychotic phenomena and ones believed to be targeted during the treatment (i.e., within the neural motor circuitry). The second specific aim is to obtain a better look at effects on the implicated regions from a functional point of view in order to gain insights about the reshaping of functional connections as a result of aerobic training, even between remote regions or areas not structurally affected. The third aim tries to combine neural plastic changes resulting from physical training with behavioral outcomes of the patient, potentially ameliorating prognosis. In addition, our last specific aim is to understand if physical treatment via promoting neural changes can be considered a potentially interesting clinical tool among the everyday rehabilitation techniques proposed for psychotic patients. 

## 2. Materials and Methods

This narrative review was conducted by following a re-adaptation of the PRISMA flow [22].

### 2.1. Inclusion Criteria

As previously stated, the aim of this narrative review was to include all research studies fulfilling the following criteria: MRI-based neuroimaging studies investigating the effects of physical training in psychotic populations. To be included, research studies had to provide enough methodological details with regard to both the MRI acquisition protocol and strategy of analysis and the type of physical training applied. No age limit was applied to the samples included as psychotic patients. Studies had to index psychotic populations via a validated measure (e.g., Positive and Negative Syndrome Scale, Structured Interview for Psychosis-risk Syndromes, Structured Clinical Interviews for DSM-5, etc.). Samples consisting of patients potentially experiencing psychosis (i.e., bipolar disorders) but lacking diagnosis and/or clinical information on the nature and specificity of the phenomena in the study manuscripts were excluded.

### 2.2. Search Strategy and Selection of Studies

In order to evaluate the relevant literature in the field, works published up to April 2023 were included in this qualitative critical review, chosen from a search conducted in open databases (PubMed, Scopus and Google Scholar). The initial search and all databases used a combination of the following terms: [physical AND (exercise OR activity), neuroimaging AND (fMRI OR DWI OR anatomy OR Voxel-based morphometry OR cortical thickness)] AND (psychosis OR schizophrenia OR psychotic disorder) without specifying a certain period of publication dates. References from the detected articles, further sources and pertinent literature reviews were also assessed for potential inclusion in the reviews. Database inclusion, exclusion, secondary searches and final inclusion, summarized in the flow diagram (Figure 1), were conducted by one of the authors (FS). 

We found 204 hits from our initial search, among which, after adjusting for duplicates, 141 articles were retained and screened for title and abstract. Additionally, 104 of these were discarded as not meeting the inclusion criteria. The full texts of the remaining 37 articles were assessed for eligibility and 18 of these papers were included in the narrative review. Moreover, the reference lists of these articles were searched for relevant records, and this search yielded 15 additional articles. Of these secondary searchers, three articles were included in the review.

Thus, out of 52 full-text screened papers, the total number of articles that met the criteria for inclusion was 21. As a last step, the entire manuscript texts of the final full list of 21 final included articles was further reassessed to ensure that they meet inclusion/exclusion criteria.

The following information was extracted from each included study: (1) authors and year of publication; (2) sample size; (3) participant characteristics (demographic information of the sample, clinical diagnosis, clinical assessment scores, cognitive scores); (4) study characteristics (design, control condition); (5) physical exercise protocol characteristics (type, frequency, duration); (6) neuroimaging study characteristics (MRI scanner, MRI sequences, outcome measure); and (7) main conclusions.

## 3. Results

In recent decades, a growing body of literature has assessed the association between neural connectivity changes and exercise intervention in patients with psychotic disorders. The search identified 21 neuroimaging studies investigating physical training effects in psychosis, 20 studies looking at structural brain plasticity and 8 at functional ones. 

These studies proposed several techniques for physical activities as treatments and investigated, by means of MRI, the neural plastic variations in brain connections both on a structural anatomical level and on a functional one. The type of physical training investigated in the above cited studies included mainly aerobic exercise (e.g., cycling, endurance training or yoga), ranging from two to three times a week, with sessions of variable duration (i.e., from 30 min up to 120 min) for an observational period of a minimum of 12 weeks. A large portion of the studies (N = 14) also included a control condition for the physical treatment frequently consisting of the absence of exercise (e.g., as waitlist or recreational/occupational activities such as table soccer). Furthermore, three of the selected studies considered only passive assessments of physical exercise without promoting physical treatment.

### 3.1. Exercise Effects on Structural Plasticity in Psychosis

#### 3.1.1. Anatomical MRI

The largest portion of selected studies (N = 19) investigated the structural modification of cortical grey matter as an effect of physical exercise (Table 1). 

Most of the studies focused on precise a priori regions where the expected structural changes (Figure 2A) should take place based rather on the pathophysiology of psychosis than supposed motor neural networking enhancement. 

Therefore, the utmost investigated brain region is the hippocampus, together with its subfields and the parahippocampal areas. Volumetric studies reported mixed results consequent to aerobic fitness, with evidences for either an increased cortical volume of the hippocampus in schizophrenic patients [25,27,30,35,39] or stable hippocampal grey matter both in clinical high-risk subjects [23,24] and later stages of psychosis [31,36,37]. The same pattern is found also while looking at the hippocampal subfields, with Damme et al. [23] reporting a negative correlation of its volume in the absence of aerobic activity and Maurus et al. [25] reporting an increased volume associated with physical activity; whereas Malchow et al. [31] describe no structural changes. Only two studies instead focused on parahippocampal regions, highlighting that in ultra-high-risk subjects for psychosis. a structural smaller volume of the area is associated with lower physical activity [34] can develop an increased white matter volume promoted by aerobic training in cases of schizophrenia [25].

Similarly, focusing on whole brain volumetric changes, other works highlighted the absence of cortical region differences in psychosis due to treatment [28,29]. Only one study exhibited subtle localized differences in temporal gyri [31] in schizophrenic patients undergoing aerobic training and also found increased volumetric grey matter in the motor and anterior cingulate cortex for the control condition (i.e., table soccer), both of which fade after physical inactivity.

Differently, as concerns effects on cortical thickness (Figure 2B), physical exercise was found to enhance prefrontal and temporal regions both in early psychosis [26,33,40] and chronic schizophrenia [38]. 

#### 3.1.2. Diffusion MRI

Only one study in the context of structural plasticity investigated the white matter structures’ modifications in schizophrenia due to physical exercise [41], showing an improvement in white matter integrity (in terms of fractional anisotropy (FA)) in motor fiber tracts thanks to training and regardless of the presence of diagnosis (i.e., both in schizophrenia and in healthy controls). Concerning patients, this structural connectivity study also corroborates previous findings about the beneficial effect of physical training on overall mental health with a peculiar alleviation of positive psychotic symptoms.

### 3.2. Exercise Effects on Functional Plasticity in Psychosis

#### Functional MRI

Looking at the total number of studies included in this narrative review, a smaller portion of the studies (N = 8) investigated the effect of exercise on functional connections in psychotic patients compared to those dedicated to structural plasticity (Table 2, Figure 2C).

Indeed, the vast majority of functional studies based their hypotheses on previous structural findings and aimed at shining new light on hippocampal changes consequent to exercise training by looking at its functional connectivity [23,24,27,32]. Functional connectivity of the hippocampus seemed to increase with the occipital lobe after physical exercise both in ultra-high risks subjects for psychosis [24] and in attenuated forms of psychosis [23] compared to control conditions. However, while looking at patients with schizophrenia, no changes were found in hippocampal functional connections to more neighboring structures (e.g., subfields, parahippocampal regions, amygdala, thalamus, middle/frontal and cingulate gyrus) after physical activity [27,32].

As regards whole-brain findings, only two studies investigated physical activity effects in psychosis on functional brain activation during tasks [28,46]. The study from Takahashi et al. is the first study to investigate functional activity changes after aerobic exercise. It demonstrated that a precise part of the occipital cortex, the extra-striate body area (EBA), increased its activity in schizophrenic patients in relation to exercise and this seemed to facilitate an improvement in psychotic symptoms. However, no further studies looked at how this increase in EBA BOLD activation relates to functional connections with other brain areas. The second task-based fMRI study, on the contrary, tried to depict a more comprehensive, connectome-based picture of brain activity changes after body-oriented interventions for psychotic patients. Their results showed a connectivity normalization of the aberrant salience network activation usually found in psychotic patients, potentially driven also by the aerobic kick-boxing training present in the body-oriented therapy. Furthermore, other functional connectivity studies highlighted an involvement of large-scale brain networking in plasticity after aerobic training, such as within the default mode network (DMN) (precuneal amplitude of low-frequency fluctuations (ALFF) decrease with yoga [45]) and the central executive network (CEN) (improved connectivity in combination with cognitive training [26]).

However, no studies have investigated the functional connectivity changes between hippocampus, DMN and SMN networking in relation to physical treatment in psychosis.

### 3.3. Behavioral Correlation with Neuroimaging Findings

A portion of the selected studies (N = 17; Table 3) concerning both structural and functional plasticity, also investigated the association of neural changes with behavioral variables that can be grouped into three main domains, namely, cognitive performance, psychotic symptomatology and social functioning.

Regarding the cognitive profile, neural changes promoted by physical exercise were found to be associated with: (i) a better general cognitive outcome in high-risk subjects [23,24]; (ii) improved short term memory in schizophrenia [29,35]; (iii) enhanced working memory and reasoning/problem solving skills both in early and overt psychosis [45]. 

Similarly, as regards behavioral clinical variables, all studies demonstrated an improved outcome as a function of exercise. Concerning clinical scores, the evidence was found for better overall psychotic symptomatology in schizophrenia [29,30,40,41,46] and for negative symptoms in early psychosis [45] and positive symptoms in high-risk cohorts [23,24]. Even though the proven effects on psychotic symptoms, there is no such clear evidence showing either an investigation of GMA or an effect of physical training in delaying the worsening/improving the management of these symptoms in association to neural changes. 

On the other hand, concerning scores of social functioning, a positive association of physical treatment with improved skills of occupation [34] and work/school [33] functioning, and social adaptation [24,38] was found in diverse conditions of psychosis.

## 4. Discussion

At the current state of the art, beside the given clinical importance of physical activity [47,48,49,50] in the context of psychosis, there is no clear consensus on the expected neural plastic changes occurring after such treatment in psychotic patients.

In this narrative review we showed the following (Figure 3): (i) hippocampal and parahippocampal structural and functional plasticity in schizophrenia and high-risk patients as a consequence of physical exercise; (ii) preliminary whole-brain functional plastic processes occurring after physical treatment and their associations with patients’ outcome.

As regards structural plasticity, most studies reported results based on a priori hypothesized regions involved in the psychotic disorder. The focus was concentrated on hippocampal and medial temporal regions, but there was no clear consensus on the volumetric effect of physical exercise across the selected studies. The only region showing consistent directionality in structural plasticity is the parahippocampus, which seemed to have a protective role both in grey and white matter portions. Prior evidence obtained from both animal and human models demonstrated that the structure and function of the parahippocampal region susceptible to physical exercise [51] are critically implicated in psychosis [52]. Reductions in parahippocampal volume have been reported in high-risk individuals with transient or isolated psychotic symptoms, and high-risk patients who developed psychosis showed longitudinal reduction. Exercise has various effects on parahippocampal regions such as increasing neural excitability, increasing grey/white matter, volumetric reduction, enhancing regional glucose metabolism, increasing cerebral blood flow, augmenting various markers of synaptic plasticity and increasing the functional connectivity with other proximal brain structures. Therefore, it is reasonable to suppose a potential role of parahippocampal regions in the motor domain. Future work could investigate the effects of exercise on connections of these regions to other areas, such as (i) the motor circuitry [13] system and (ii) or temporal and frontal regions, which were previously found to be structurally affected by aerobic training in psychosis [31,38].

Concerning structural connections changes, we also highlighted that only one study investigated the effect of exercise on white matter motor bundles. Brain white matter alterations are well-documented in adults with psychotic disorders; the results indicate widespread lower FA, with the largest effect sizes in the anterior corona radiata and corpus callosum (CC). The involvement of the CC seems to be connected to the role of abnormal hemispheric specialization and abnormal interhemispheric communication in the etiology of the disease. Accumulating evidence suggests that aerobic exercise training and higher cardiorespiratory fitness are associated with improved microstructural organization of the CC; in particular, practicing aerobic exercise regularly or intense cardiorespiratory fitness leads to an increase in FA values in the CC. Despite the idea that temporary physical training might not be persistent or effective enough to provoke such structural changes, all these elements highlight the need of a more extensive investigation of structural connectivity.

In addition, a small number of studies looked at functional plasticity reporting based on regions structurally affected by exercise, changes in connectivity of the hippocampus and in large scale networking. However, the generalizability of these results is very poor due to power issues limiting the possibility to draw comprehensive conclusions; hence, further studies are needed. 

Moreover, the selected studies investigating behavioral associations with neural changes due to physical treatment consistently reported a positive effect on multiple domains affected by the presence of psychosis (i.e., cognitive deficits, psychotic symptomatology and functional outcome). This poses aerobic exercise as a potentially effective treatment for schizophrenia and psychotic spectrum disorders in general, as previously hypothesized [48]. However, the level of persistence of plastic neural changes over time after the conclusion of the treatment, which is thought to re-organize the psychotic brain back to physiology, is still a matter of debate. 

Nevertheless, there is another relevant point to raise regarding the investigation of motor circuitry changes. Previous investigations [13] demonstrated how three main networks involved in motor functions, namely, the basal ganglia, the cerebellar–thalamocortical and the corticomotor circuits, are relevant for the pathophysiology of psychosis. With this review, we have highlighted the lack of studies investigating how these connections can be affected or influenced by physical and motor exercise in psychosis. Indeed, the vast majority of studies included in this narrative review were focused mainly on brain changes occurring after physical treatment in regions allegedly expected to be affected [8] by psychosis symptomatology, rather than on motor connections. Furthermore, in this narrative review no evidence was found investigating the correlation between the rise in neural plasticity thanks to exercise and the effect on GMA symptoms, leaving the door open for upcoming investigations. Therefore, future experimental studies should examine changes in the inter-relationships of motor networks in psychosis as a result of aerobic exercise to better probe its potential as a targeted treatment on GMA.

Most of the studies focus on the effects of aerobic activity, although there is no literature that guarantees a better effect of this physical exercise over others. This choice is probably due to a greater simplicity in conducting the intervention compared to team sports or other sports such as contact sports, which could have a positive impact on the dopaminergic circuits but increase aggressive attitudes. It would be of interest to compare various modalities of exercise with other forms of intervention, including medication, cognitive behavioral therapy and imagined locomotion. In future studies, it will also be important to acquire information on the amount of previously played sports to correlate with the MRI acquired before treatment, as the outcome of the therapy could be different in relation to patient’s habit of physical exercise.

However, the limitations of this study must be considered. Being a narrative review, it was beyond the scope of this work to systematically investigate the quantitative effects of physical treatment in psychosis; therefore, future meta-analyses could shed light on precise estimated effects.

Despite this limitation, this review is of scientific interest as it underlines how physical exercise can have a strong therapeutic impact on those suffering from psychosis.

## 5. Conclusions

In this review, we provide evidence of both structural and functional brain plasticity in psychosis as a result of physical treatment. Although there is no obvious directionality and the effects do not last once the exercise phase is over, physical training, especially aerobic exercise, promotes structural changes in the areas of the hippocampus. Physical exercise and brain changes altered the hippocampus’ functional connectivity as well as that of other large-scale networks, and these changes were associated with improved functional outcomes for patients. To evaluate the long-term effectiveness of physical therapy within the psychotic spectrum and its reliability in the therapeutic setting, more research is still required. Future studies should concentrate on enhancing our understanding of how neural connections within motor circuitry alter in psychosis, emphasizing both cortical and subcortical connection changes.

## Figures and Tables

**Figure 1 brainsci-13-00923-f001:**
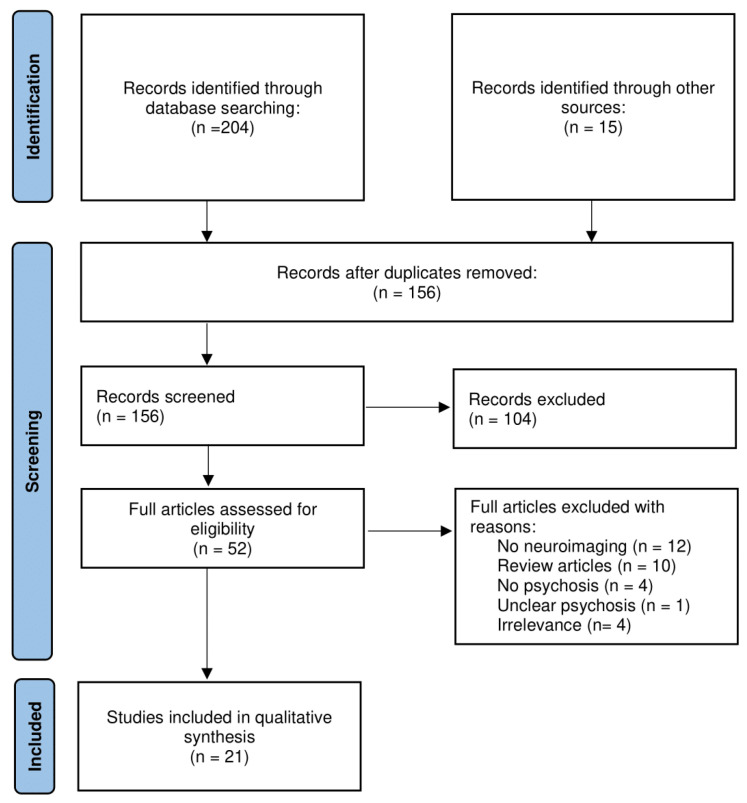
Flowchart of the narrative review process.

**Figure 2 brainsci-13-00923-f002:**
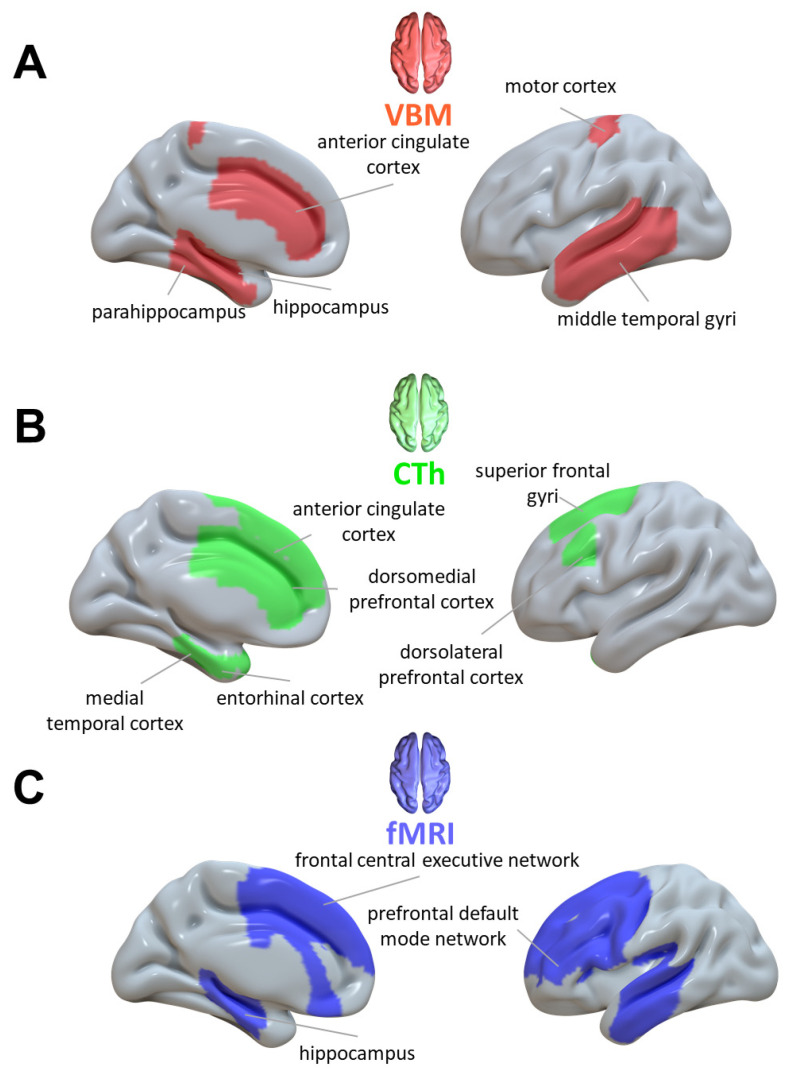
Neural plasticity processes. Evidence from neuroimaging studies on (**A**) structural morphometric (Red: VBM), (**B**) structural cortical thinning (Green: CTh) and (**C**) functional (Blue: fMRI) plasticity changes occurring in brain regions of psychotic patients as a function of physical activity. The figure was generated with Surf Ice toolbox (https://www.nitrc.org/projects/surfice/) (accessed on 29 April 2023)by rendering on the surface mesh previously selected and atlas-derived regions of interest (Harvard–Oxford cortical and subcortical structural atlases [42]; Anatomical labelling Atlas [43] (AAL3); Atlas of Intrinsic Connectivity of Homotopic Areas [44] (AICHA)) found to be implicated in previous studies. Abbreviations: VBM (voxel-based morphometry); CTh (cortical thickness); fMRI (functional MRI).

**Figure 3 brainsci-13-00923-f003:**
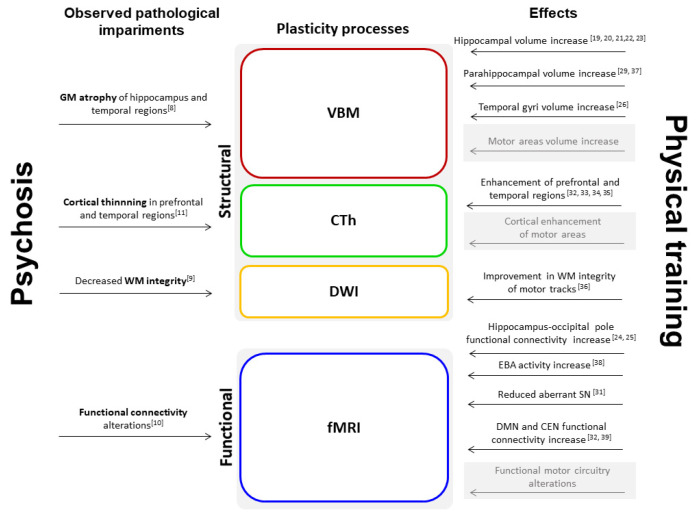
Schematic representation of physical training effects on multiple levels and relevant observed pathological impairments in psychosis. Observed effects are reported with the relevant reference, whereas the expected effects still lacking investigations are reported in grey. Abbreviations: VBM (voxel-based morphometry); CTh (cortical thickness); DWI (diffusion-weighted imaging); fMRI (functional MRI); GM (grey matter); WM (white matter); EBA (extra-striate body area); SN (salience network); DMN (default mode network); CEN (central executive network).

**Table 1 brainsci-13-00923-t001:** Structural plasticity studies: research articles investigating neural structural plasticity changes due to the effect of physical exercise in psychosis.

Author	Participant Characteristics						Study Characteristics		Exercise Protocol Characteristics			Neuroimaging Characteristics			Outcomes
	N	Age (Mean)	Gender (F)	Diagnosis	Clinical Scores	Cognitive Scores	Design	Control Condition	Exercise	Frequency	Duration	Scanner	Sequence	Outcome Measures	
Damme et al., 2022 [23]	25	21	11	CHR	SIPS	WRAT, RISE	RCT	Waitlist	AE	2/week	3 months	3T	T1-MPRAGE	HP and subfields GMV	(1) Stable HP volumes; (2) Decreased HP subfield volume with no exercise.
Dean et al., 2017 [24]	12	19.4	6	UHR	SIPS, GFS	MCCB	RCT	None	AE	3–2/week	12 weeks	3T	T1-MPRAGE	HP GMV	(1) No changes in HP volume.
Maurus et al., 2022a [25]	48	37.4	19	SCZ	N.A.	VLMT	CS	N.A.	AE assessment	N.A.	N.A.	3T	T1-MPRAGE	GMV of HP and subfields	(1) Positive associations of aerobic fitness levels and HP subfields volumes.
McEwen et al., 2017 [26]	37	N.A.	N.A.	FEP	N.A.	MCCB	LS	CT	CT and AE	N.A.	N.A.	N.A.	N.D.	CTh	(1) Increased CTh were in prefrontal regions.
Roell et al., 2023 [27]	92	36.8	31	SCZ	PANSS, GAF, FROGS	VLMT, Digit Span Test, TMT	RCT	NA-T	AE	N.A.	3 months	3T	T1-MPRAGE	GMV of HP and subfields	(1) In the AE group volumes within the HP formation increased.
van der Stouwe et al., 2021 [28]	31	34.3	12	PSY	PANSS, BNSS	N.A.	LRCT	Befriending	BEATVIC	1/week (75 min)	20 sessions	3T	T1-w image	VBM	(1) No differences at VBM.
Falkai et al., 2013 [29]	16	35.2	16	SCZ	PANSS	VLMT, Corsi block Tapping test	RCT	Table football	AE (cycling)	3/week	3 months	1.5 T	T1-MPRAGE; T2-w gradient echo	GMD, CSE	(1) Cortical changes in healthy controls; (2) No effect in cortical regions for SCZ.
Lin et al., 2015 [30]	124	24.5	124	SCZ, SCZA, PSY	PANSS, CDSS, QoL, FRS, CRS	VA, VR, Digit Span Test, LC Q score, Stroop task	RCT	Waitlist	Yoga or AE	3/week	12 weeks	3T	T1-MPRAGE	HP GMV	(1) AE increases HP volume.
Malchow et al., 2016 [31]	20	35.8	12	SCZ	PANSS	N.A.	LS	Table football	Endurance training	3/week	3 months	3T	T1-MPRAGE	VBM, manual and automatic segmentation of HP and subfields.	(1) No volume increase in HP and its subfields; (2) Endurance training increased volume of the left superior, middle and inferior anterior temporal gyri; (3) Table soccer increased volumes in the motor and anterior cingulate cortices; (4) Differences were no longer present after inactivity.
Maurus et al., 2022b [32]	69	36.94	23	SCZ	PANSS, CDSS	TMT, Digit Span Test, VLMT, B-CATS, DSST, ERT	CS	N.A.	AE assessment	N.A.	N.A.	3T	T1-MPRAGE	Anatomical parcellation for GMV and WMV	(1) AE is associated to increased right HP GMV and para-HP WMV.
McEwen et al., 2023 [33]	37	22.7	10	FEP	N.A.	MCCB	LRCT	CT (Brain HQ)	CT + AE	3/week	24 weeks	3T	T1-MPRAGE; T2-weighted turbo spin-echo	CTh	(1) CT and AE increase CTh within the left anterior cingulate cortex over treatment period; (2) Directional tendencies were similar in the left dorsolateral prefrontal cortex.
Mittal et al., 2013 [34]	29	18.5	11	UHR	SIPS, SCID	N.A.	CS	N.A.	wristwatch recording (ActiLife scoring)	N.A.	5 days	3T	T1-MPRAGE; T2-weighted turbo spin-echo	GMV of HP and para-HP gyri	(1) UHR greater percentage of time in sedentary behavior; (2) Trend UHR group showed less total physical activity; (3) UHR smaller medial temporal volumes; (4) Inactivity is associated with medial temporal lobe health.
Pajonk et al., 2010 [35]	16	35.2	0	SCZ	PANSS, CGI	VLMT, Corsi block Tapping Test	RCT	Table football	AE (cycling)	3/week	3 months	1.5T	T1-MPRAGE; T2-weighted gradient echo; spin-echo MRS	GMV HP; neuro-metabolites of left HP	(1) AE increase HP volume in patients with no change in the non-exercise group of patients; (2) HP volume in the exercise group were correlated with improvements in aerobic fitness; (3) SCZ exercise group changes in HP volume were associated with a 35% increase in NAA/Cr in HP;
Rosenbau et al., 2015 [36]	5	N.A.	0	SCZ, SCZA	SAPS, SANS, WHOQOL-BREF	VLMT, WMS	Pilot	None	Stationary bike	2/week	12 weeks	1.5T	T1-MPRAGE	GMV of HP	(1) No significant HP volume increase.
Scheewe et al., 2013 [37]	32	29.8	6	SCZ	PANSS	WAIS IQ	RCT	OC	AE	1/week	6 months	3T	T1-weighted FFE	GMV, CTh	(1) AE does not affect global brain and HP volume or CTh in patients and controls; (2) CRF improvement was related to increased cerebral matter volume and lateral and third ventricle volume decrease in patients and to thickening in the left hemisphere in large areas of the frontal, temporal and cingulate cortex irrespective of diagnosis; (3) One to two hours’ exercise therapy did not elicit significant brain volume changes in patients or controls.
Takahashi et al., 2020 [38]	21	36.2	12	SCZ	GAF, SAS, PANSS, CGI, CDSS	VLMT, TMT, WCST	LS	Table soccer	AE (cycling)	1/week	12 weeks	3T	T1-MPRAGE	CTh of entorhinal, para-HP and lateral/medial PFC	(1) AE showed increase in CTh in right entorhinal cortex; (2) No significant longitudinal change in CTh in control groups.
Woodward et al., 2018 [39]	17	30.1	11	SCZ, SCZA	PANSS	WAR	LRCT	Weight-bearing exercise program	AE	3/week	12 weeks	3T	T1-MPRAGE; SWI	GMV HP and its subfields; SWI mapping of HP vasculature volume	(1) HP increased in the left CA-1 field; (2) HP vascular volume unchanged.
Woodward et al., 2020 [40]	51	24.5	51	FEP	PANSS	N.A.	RCT	Waitlist	AE/ Hatha yoga	3/week	12 weeks	3T	T1-MPRAGE	GMV and CTh of whole-brain and HP	(1) Increases in GMV and CTh in the medial temporal cortical regions for AE.
Svatkova et al., 2015 [41]	33	30	6	SCZ	PANSS	IQ	LS	Life-as-usual	AE + NAE	1/week	6 months	3T	DWI	DTI and TBSS of major tracts	(1) Physical exercise increases the integrity of white matter fiber tracts; (2) Life-as-usual decreases fiber integrity.

Abbreviations: N.A. (not applicable); N.D. (not defined); N (number of participants); F (female); CHR (presence of diagnosed attenuated psychosis syndrome); UHR (ultra high-risk for psychosis); SCZ (schizophrenia); FEP (first-episode psychosis); PSY (psychotic spectrum); SCZA (schizoaffective disorder); SIPS (Structured Interview for Psychosis-risk Syndromes); GFS (global level of functioning); PANSS (Positive and Negative Syndrome Scale); GAF (Global Assessment of Functioning,); FROGS (Functional Remission of General Schizophrenia); BNSS (Brief Negative Symptom Scale); CDSS (Calgary Depression Scale for Schizophrenia); QoL (Quality of Life Scale); FRS (first-rank symptoms); CRS (Compliance Rating Scale); SCID (Structured Clinical Interview for DSM Disorders), CGI (Clinical Global Impression); SAPS (Scale for the Assessment of Positive Symptoms); SANS( Scale for the Assessment of Negative Symptoms); SAS (Self-Rating Anxiety Scale); WHOQOL-BREF (World Health Organization Quality of-Life Scale); WRAT (Wide Range Achievement Test); RISE (Relational and Item-Specific Encoding Task); MCCB (MATRICS Consensus Cognitive Battery); VLMT (Verbal learning and Memory test); TMT (Trial Making Test); VA (verbal acquisition); VR (verbal retention); LC Q score (Letter Cancellation Test); B-CATS (The Brief Cognitive Assessment Tool for Schizophrenia); DSST (Digit Symbol Substitution Test); ERT (Emotion Recognition Test); WMS (Weschler Memory Scale); WAIS IQ (Wechsler Adult Intelligence Scale); WCST (Wisconsin Card Sorting Test); WAR (Weschler Test of Adult Reading); IQ (intelligence quotient); RCT (randomized controlled trial); LRCT (longitudinal randomized controlled trial); LS (longitudinal study); CS (cross-sectional study); CT (cognitive training); NA-T (flexibility, strengthening and balance training non aerobic); AE (aerobic exercise); NAE (non-aerobic exercise); BEATVIC (session led by a therapist trained in body and movement-oriented interventions); T1-MEMPRAGE (T1-Multi-echo-MPRAGE); MRS (magnetic resonance spectroscopy); SWI (susceptibly weighted imaging); DWI (diffusion-weighted imaging); HP (hippocampus); GMV (grey matter volume); CTh (cortical thickness); VBM (voxel-based morphometry); GMD (grey matter density); CSE (cortical surface expansion); WMV (white matter volume); PFC (prefrontal cortex); DTI (diffusion tensor imaging); TBSS (tract-based spatial statistics); N-acetylaspartate-to-creatine ratio (NAA/Cr); CRF (cardiorespiratory fitness).

**Table 2 brainsci-13-00923-t002:** Functional plasticity studies: research articles investigating neural functional plasticity changes due to the effect of physical exercise in psychosis.

Author	Participant Characteristics						Study Characteristics		Exercise Protocol Characteristics			Neuroimaging Characteristics			Outcomes
	N	Age (Mean)	Gender (F)	Diagnosis	Clinical Scores	Cognitive Scores	Design	Control Condition	Exercise	Frequency	Duration	Scanner	Sequence	Outcome Measures	
Damme et al., 2022 [23]	25	21	11	CHR	SIPS	WRAT, RISE	RCT	Waitlist	AE	2/week	3 months	3T	T1-MPRAGE; rs-fMRI	ROI-ROI FC (HP-occipital lobe)	(1) Increased HP connectivity.
Dean et al., 2017 [24]	12	19.4	6	UHR	SIPS, GFS	MCCB	RCT	None	AE	3–2/week	12 weeks	3T	T1-MPRAGE; rs-fMRI	ROI-ROI FC (left HP—right HP—bilateral occipital cortices)	(1) Increased FC between left HP and occipital cortex.
Lin et al., 2018 [45]	124	N.A.	124	FEP	PANSS	Working memory	RCT	Waitlist	Yoga/AE	N.A.	12 weeks	N.A.	N.D.	ALFF	(1) ALFF decreases in precuneus in yoga group and correlates with better negative symptoms.
Maurus et al., 2022a [25]	48	37.4	19	SCZ	N.A.	VLMT	CS	N.A.	AE assessment	N.A.	N.A.	3T	T1-MPRAGE; rs-fMRI	FC matrices of HP and subfields	(1) No associations of HP subfields FC or mediation effects on verbal memory.
McEwen et al., 2017 [26]	37	N.A.	N.A.	FEP	N.A.	MCCB	LS	CT	CT and AE	N.A.	N.A.	N.A.	N.D.	FC	(1) CT and exercise improved FC between right CEN and ventral attention network and also between left CEN and right CEN.
Roell et al., 2023 [27]	92	36.8	31	SCZ	PANSS, GAF. FROGS	VLMT, Digit Span Test, TMT	RCT	NA-T	AE	N.A.	3 months	3T	T1-MPRAGE; rs-fMRI	FC from the HP subfields, the striatum, the amygdala and thalamus, the DLPFC and CC	(1) No effects of exercise on HP formation connectivity were observed.
Takahashi et al., 2012 [46]	23	41.7	11	SCZ	PANSS	N.A.	LS	HC	AE	2/day (60–120 min)	3 months	1.5 T	T1-w image; Task-fMRI	Task-based fMRI GLM	(1) Activation of the body-selective EBA in the posterior temporal-occipital cortex during observation of sports-related actions was increased in the program group
van der Stouwe et al., 2021 [28]	31	34.3	12	PSY	PANSS, BNSS	N.A.	LRCT	Befriending	BEATVIC	1/week (75 min)	20 sessions	3T	T1-weighted image; Task-fMRI	Task-based fMRI GLM and ICA FC	(1) GLM no differences between groups over time; (2) ICA increased activation of the salience network to angry and fearful faces in BEATVIC; (3) Increased activation of the salience network may suggest an increased alertness for potentiallydangerous faces.

Abbreviations: N.A. (not applicable); N.D. (not defined); N (number of participants); F (female); CHR (presence of diagnosed attenuated psychosis syndrome); UHR (ultra high-risk for psychosis); SCZ (schizophrenia); FEP (first-episode psychosis); PSY (psychotic spectrum); SCZA (schizoaffective disorder); SIPS (Structured Interview for Psychosis-risk Syndromes); GFS (global level of functioning); PANSS (Positive and Negative Syndrome Scale); GAF (Global Assessment of Functioning;); FROGS (Functional Remission of General Schizophrenia); BNSS (Brief Negative Symptom Scale); WRAT (Wide Range Achievement Test); RISE (Relational and Item-Specific Encoding Task); MCCB (MATRICS Consensus Cognitive Battery); VLMT (Verbal learning and Memory test; TMT (Trial Making Test); RCT (randomized controlled trial); LRCT (longitudinal randomized controlled trial); LS (longitudinal study); CS (cross-sectional study); CT (cognitive training); NA-T (flexibility, strengthening and balance training non aerobic); HC (healthy controls); Befriending (social interaction group intervention); AE (aerobic exercise); NAE (non-aerobic exercise); BEATVIC (session led by a therapist trained in body and movement-oriented interventions); rs-fMRI (resting-state fMRI); ROI (region of interest); HP (hippocampus); ALFF (amplitude of low-frequency fluctuations); FC (functional connectivity); DLPFC (dorsolateral prefrontal cortex); CC (corpus callosum); GLM (general linear model); ICA (independent component analysis); CEN (central executive network); EBA (extra-striate body Area).

**Table 3 brainsci-13-00923-t003:** Behavioral correlations in neuroimaging studies: research articles investigating how neural plastic changes due to the effect of physical exercise in psychosis are correlated with behavioral/cognitive and clinical outcomes.

Author	Participant Characteristics						Study Characteristics		Exercise Protocol Characteristics			
	N	Age (Mean)	Gender (F)	Diagnosis	Clinical Scores	Cognitive Scores	Design	Control Condition	Exercise	Frequency	Duration	
Damme et al., 2022 [23]	25	21	11	CHR	SIPS	WRAT, RISE	RCT	Waitlist	AE	2/week	3 months	(1) Improved fitness; (2) Increased cognitive performance; (3) Decrease in positive symptoms.
Dean et al., 2017 [24]	12	19.4	6	UHR	SIPS, GFS	MCCB	RCT	None	AE	3–2/week	12 weeks	(1) Improved positive and negative symptoms; (2) Improved social functioning; (3) Improved cognition.
Maurus et al., 2022a [25]	48	37.4	19	SCZ	N.A.	VLMT	CS	N.A.	AE assessment	N.A.	N.A.	(1) No associations of HP subfields FC or mediation effects on verbal memory.
McEwen et al., 2017 [26]	37	N.A.	N.A.	FEP	N.A.	MCCB	LS	CT	CT and AE	N.A.	N.A.	(1) Improved FC between left and right CEN was associated with cognitive gains in reasoning and problem solving at 6-month follow-up.
Falkai et al., 2013 [29]	16	35.2	16	SCZ	PANSS	VLMT, Corsi block Tapping test	RCT	Table football	AE (cycling)	3/week	3 months	(1) Improved short-term memory; (2) Improved PANSS.
Lin et al., 2015 [30]	124	24.5	124	SCZ, SCZA, PSY	PANSS, CDSS, QoL, FRS, CRS	VA, VR, Digit Span Test, LC Q score, Stroop task	RCT	Waitlist	Yoga or AE	3/week	12 weeks	(1) Yoga and AE improved working memory; (2) Yoga improved verbal acquisition and attention; (3) Yoga and AE improved overall and depressive symptoms;
Malchow et al., 2016 [31]	20	35.8	12	SCZ	PANSS	N.A.	LS	Table football	Endurance training	3/week	3 months	(1) Psychopathological symptoms did not change.
Maurus et al., 2022b [32]	69	36.94	23	SCZ	PANSS, CDSS	TMT, Digit Span Test, VLMT, B-CATS, DSST, ERT	CS	N.A.	AE assessment	N.A.	N.A.	(1) No association between cognition or symptoms and AE.
McEwen et al., 2023 [33]	37	22.7	10	FEP	N.A.	MCCB	LRCT	CT (Brain HQ)	CT + AE	3/week	24 weeks	(1) Increased CTh in the left ACC was improved work/school functioning.
Mittal et al., 2013 [34]	29	18.5	11	UHR	SIPS, SCID	N.A.	CS	N.A.	wristwatch recording (ActiLife scoring)	N.A.	5 days	(1) Total level of physical activity in UHR correlated with smaller para-HP gyri bilaterally and with occupational functioning.
Pajonk et al., 2010 [35]	16	35.2	0	SCZ	PANSS, CGI	VLMT, Corsi block Tapping Test	RCT	Table football	AE (cycling)	3/week	3 months	(1) Short-term memory improvement in SCZ was correlated with change in HP volume.
Takahashi et al., 2020 [38]	21	36.2	12	SCZ	GAF, SAS, PANSS, CGI, CDSS	VLMT, TMT, WCST	LS	Table soccer	AE (cycling)	1/week	12 weeks	(1) Significant correlation between CTh of right lateral PFC at baseline and improvement of social adaptation
Woodward et al., 2018 [39]	17	30.1	11	SCZ, SCZA	PANSS	WAR	LRCT	weight-bearing exercise program	AE	3/week	12 weeks	(1) Changes in HP volume and vascular volume were not significantly correlated with changes in symptom severity, nor did they affect scores.
Woodward et al., 2020 [40]	51	24.5	51	FEP	PANSS	N.A.	RCT	Waitlist	AE/ Hatha yoga	3/week	12 weeks	(1) AE increases in the entorhinal and fusiform/temporal gyri associated with reduced symptom severity; (2) Increased fusiform CTh associated with increased HP volume for all psychosis participants.
Svatkova et al., 2015 [41]	33	30	6	SCZ	PANSS	IQ	LS	Life-as-usual	AE + NAE	1/week	6 months	(1) Exercise improves brain structural connectivity and positive symptoms.
Lin et al., 2018 [45]	124	N.A.	124	FEP	PANSS	Working memory	RCT	Waitlist	Yoga/AE	N.A.	12 weeks	(1) Yoga and aerobic exercise improved working memory and psychotic symptoms; (2) ALFF decreases in precuneus in yoga group and correlates with better negative symptoms.
Takahashi et al., 2012 [46]	23	41.7	11	SCZ	PANSS	N.A.	LS	HC	AE	2/day (60–120 min)	3 months	(1) Increase in EBA activation was associated with PANSS improvement.

Abbreviations: N.A. (not applicable); N.D. (not defined); N (number of participants); F (female); CHR (presence of diagnosed attenuated psychosis syndrome); UHR (ultra high-risk for psychosis); SCZ (schizophrenia); FEP (first-episode Psychosis); PSY (psychotic spectrum); SCZA (schizoaffective disorder); SIPS (Structured Interview for Psychosis-risk Syndromes); GFS (global level of functioning); PANSS (Positive and Negative Syndrome Scale); GAF (Global Assessment of Functioning;); FROGS (Functional Remission of General Schizophrenia); BNSS (Brief Negative Symptom Scale); CDSS (Calgary Depression Scale for Schizophrenia); QoL (Quality of Life Scale); FRS (First Rank Symptoms); CRS (Compliance Rating Scale); SCID (Structured Clinical Interview for DSM Disorders); CGI (Clinical Global Impression); SAPS (Scale for the Assessment of Positive Symptoms); SANS (Scale for the Assessment of Negative Symptoms); SAS (Self-Rating Anxiety Scale); WHOQOL-BREF (World Health Organization Quality of Life Scale); WRAT (Wide Range Achievement Test); RISE (Relational and Item-Specific Encoding Task); MCCB (MATRICS Consensus Cognitive Battery); VLMT (Verbal learning and Memory test); TMT (Trial Making Test); VA (verbal acquisition); VR (verbal retention); LC Q score (Letter Cancellation Test); B-CATS (The Brief Cognitive Assessment Tool for Schizophrenia); DSST (Digit Symbol Substitution Test); ERT (Emotion Recognition Rest); WMS (Weschler Memory Scale); WAIS IQ (Wechsler Adult Intelligence Scale); WCST (Wisconsin Card Sorting Test); WAR (Weschler Test of Adult Reading); IQ (intelligence quotient); RCT (randomized controlled trial); LRCT (longitudinal randomized controlled trial); LS (longitudinal study; CS (cross-sectional study); CT (cognitive training); NA-T (flexibility, strengthening and balance training non aerobic); AE (aerobic exercise); NAE (non-aerobic exercise); BEATVIC (session led by a therapist trained in body and movement-oriented interventions); HP (hippocampus); FC (functional connectivity); CEN (central executive network); CTh (cortical thickness); ACC (anterior cingulate cortex); PFC (prefrontal cortex); EBA (extra-striate body area); ALFF (amplitude of low frequency fluctuations).

## Data Availability

Not applicable.

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
