# Peer review of "The Effect of Physical Exercise on People with Psychosis: A Qualitative Critical Review of Neuroimaging Findings"

_brainsci, 2023, doi:10.3390/brainsci13060923_

Round 1
Reviewer 1 Report
Major concerns:
1. As a review article, the author should provide an intricate and extensive account of the paper selection process, incorporating the databases and keywords utilized to conduct the literature searches, the criteria used for excluding studies, and other crucial details. By failing to provide such information, readers may not acquire a precise insight into the comprehensiveness of the literature review and the authentic nature of the presented data. Addressing these concerns necessitates a more comprehensive explanation from the authors regarding the paper selection process, which would enhance the robustness and transparency of the research.
2. The discussion section of the article fails to provide in-depth analysis of the research findings and instead offers only a brief summary of previous sections. To improve the quality of the study, the authors must strive to strengthen the discussion section by investigating the potential neurophysiological and biological mechanisms that underpin the outcomes presented in the preceding sections. Thorough exploration of the neurobiological basis of the findings would effectively reinforce the study’s reliability and significance.
English is ok
Author Response
We thank the reviewers for their helpful comments and suggestions. Here we outline our responses and indicate where in the manuscript we made relevant revisions by highlighting in yellow the changes.

Reviewer 2 Report
This is a review focused on the effect of physical exercise in people suffering from psychosis. The paper is well written and of interest for the journal; however, several changes in the methods and discussion should be made.
ABSTRACT
1- The abstract is really short. I recommend to expand the aims, and methods of the review. A conclusion is needed, without summarizing results.
INTRODUCTION
1- The introduction should be expanded by describing several interventions that can be addressed in patients suffering from psychosis. I would focus on non-pharmacological interventions. In this section, the authors should emphasize why is important to focus on physical exercise.
2- The main aims of the paper should be expanded and included in a separate section (1.1. Aims).
MATERIALS AND METHODS
This section is really brief. I recommend to describe the screening and selection processes, the design of the review (narrative, systematic, etc).
A flow chart is needed to explain why papers were excluded or included. Inclusion and exclusion criteria are really important.
RESULTS
Table 1. The title and Journal is not needed in a Table. If papers are referenced, this information can be obtained via the list of references (at the end of the manuscript).
Tables 2 and 3 require citations according to the journal style "[ ]".
DISCUSSION
The discussion is brief. I recommend to expand this section by adding, also a subsection about Future perspective. Is there any specific recommendation of physical exercise?
CONCLUSIONS
The conclusions should not include the limitations or strenghts, and should not be a summarize of the main results.
Author Response

(The authors gave the same response as above.)

Reviewer 3 Report
According to the concept of genuine motor abnormalities (GMA), people with psychosis exhibit patterns of physical characteristics early (prodromal). Early attempts to alleviate such patterns have been reported, e.g. a period of physical exercises. The authors critically review brain changes associated with the effect of physical exercises in people with psychosis. The authors wrote some evidence from past studies of both structural and functional plasticity and discussed regions previously thought to be involved in the pathology of the diseases.
Please note you need to revise the following, especially in the methodology part:
1) Abstract:
The writing is too thin. Add 2-3 sentences, by summarizing what brain regions are involved based on the past literature. Also, check Line 12 as it sounds off: “… clinical tool if promoting..”
2) Materials: search from past literature
The search strategy is lacking. I invite the authors to adopt the PRISMA flow and see if they have previously followed this while producing the research.
I am unsure if you are aiming at general psychosis or focusing on schizophrenia. Is bipolar disorder excluded? Psychotic disorders comprise a few different types.
Usually the authors mentioned how many studies were found by the keywords, how many are filtered out and subsequently included as the primary source of information. You did not specify the process leading to the individual studies which reported functional MRI, structural, and so on. Which of the authors performed the search, did the reading, and synthesizing the information. Such information shall be shared.
3) Results & Discussion:
How Figure 1 was generated deserves more explanation. Is it a reproduction of past work? What software tool did you use to produce it?
Also, can the authors clarify the context of physical exercise in this study? If the manuscript begins by discussing the GMA which appears usually early, are those 21 articles discussed here also targeting the GMA? Will physical exercise have the effect of delaying the worsening (early intervention) or managing the overt symptoms? The authors can link back the narrative of GMA in the context of psychosis.
In general okay!
Author Response

(The authors gave the same response as above.)

Round 2
Reviewer 1 Report
The problems previously identified have been appropriately modified, greatly improving the overall structure and readability of the manuscript.
Reviewer 2 Report
The authors have done a great work by considering all the comments we suggested. The paper has been successfully improved.
Reviewer 3 Report
The authors have adequately addressed the concerns.
Thank you.